# Wireless Body Area Sensor Networks: Survey of MAC and Routing Protocols for Patient Monitoring under IEEE 802.15.4 and IEEE 802.15.6

**DOI:** 10.3390/s22218279

**Published:** 2022-10-28

**Authors:** Muhammad Sajjad Akbar, Zawar Hussain, Michael Sheng, Rajan Shankaran

**Affiliations:** 1School of Computing, Macquarie University Australia, Sydney, NSW 2109, Australia; 2School of Computer Science and Engineering, The University of New South Wales Australia, Sydney, NSW 2052, Australia

**Keywords:** WBASN, patient monitoring systems, IoTs, RSSI, LQI, link quality, routing, IEEE 802.15.4, IEEE 802.15.6

## Abstract

Wireless body area sensor networks (WBASNs) have received growing attention from industry and academia due to their exceptional potential for patient monitoring systems that are equipped with low-power wearable and implantable biomedical sensors under communications standards such as IEEE 802.15.4-2015 and IEEE 802.15.6-2012. The goal of WBASNs is to enhance the capabilities of wireless patient monitoring systems in terms of data accuracy, reliability, routing, channel access, and the data communication of sensors within, on and around the human body. The huge scope of challenges related to WBASNs has led to various research publications and industrial experiments. In this paper, a survey is conducted for the recent state-of-art in the context of medium access control (MAC) and routing protocols by considering the application requirements of patient monitoring systems. Moreover, we discuss the open issues, lessons learned, and challenges for these layers to provide a source of motivation for the upcoming design and development in the domain of WBASNs. This survey will be highly useful for the 6th generation (6G) networks; it is expected that 6G will provide efficient and ubiquitous connectivity to a huge number of IoT devices, and most of them will be sensor-based. This survey will further clarify the QoS requirement part of the 6G networks in terms of sensor-based IoT.

## 1. Introduction

The major healthcare challenges for the world population are the growth of the elderly population due to improved life expectancy, the rise in healthcare costs, and the high death rate because of chronic diseases [1]. It is estimated that high growth in the aging population will overload the healthcare systems; moreover, it is expected that remote healthcare monitoring services will be required for a population of 761 million people in 2025 [2]. The growth of the elderly population in developed countries and their healthcare cost have triggered technology-driven enhancements to current healthcare practices. For instance, recent advances in electronics have enabled the development of intelligent body monitoring sensors that are wearable on the body and implantable in the human body, as shown in Figure 1. These sensors send their data to a distributed server to store and analyse the data. Sensor nodes are capable of reliable data transmission, and the collected information must be sent to the coordinator node. However, there is a trade-off between reliability, delay and energy consumption [3,4,5]. Hence, while designing the MAC and routing protocols, WBASN protocol designers should consider these trade-offs. Due to the lack of battery replacement options for these sensors, there is a need to design energy-efficient MAC and routing protocols; moreover, less delay and high reliability tend to consume more energy [6,7,8,9]. Hence, the minimisation of delay and maximisation of reliability in WBASNs is not considered an optimal design approach unless they are designed by considering the lifetime of sensor nodes. Despite the limited battery power of body sensors, some devices are required to work unobtrusively for months or even years. Further, in MAC protocols, energy waste is due to idle listening, collision, packet overhead, and overhearing. WBASNs consist of multiple physiological sensors that require different data rates, as shown in Table 1. Therefore, the selection of an appropriate radio frequency (RF) is a crucial part of deploying patient monitoring systems. The heterogeneous nature of biomedical sensors in terms of sensing and transmitting data make the required QoS more complex for the MAC layer as it may now need to send some data with high priority, such as electrocardiography (ECG) data in an emergency scenario. For such communication, using the wired solution will be expensive. However, the use of wireless infrastructure is more cost-efficient and easily deployable. Wireless patient monitoring provides flexibility in terms of mobility; further, it helps to remotely monitor elderly people.

Every year, millions of people suffer from diseases, including cardiovascular, blood pressure, asthma, diabetes, etc. Research has revealed that most of these diseases can be avoided if they are identified early. In this context, patient monitoring systems [10,11,12,13,14,15,16,17,18,19,20,21,22,23,24] could be very useful in providing quality of life without disrupting the patients’ daily routines. WBASNs integrate with other communication technologies, including ZigBee, Bluetooth, wireless local area networks (WLANs) and wireless personal area networks (WPANs). Usually, for communications at the MAC and physical layers, WBASNs use IEEE 802.15.4 low-rate wireless personal area networks (LR-WPANs) and IEEE 802.15.6 wireless body area networks (WBANs). ZigBee is a popular industrial standard that works above the IEEE 802.15.4 MAC layer and is widely available in the market as a ready product [1].

Various challenges have been discussed when using different methodologies and approaches [25,26,27,28,29,30,31,32,33,34]. The QoS concerns in WBASNs are different and more challenging as the patient monitoring applications generate a large number of small packets in short time intervals; these packets require timely and reliable delivery with efficient channel access. The collected data must reach the sink (coordinator) node within a predefined threshold of delay; otherwise, it will not remain meaningful [16,17,18]. In WBASNs, latency must be less than 250 ms for most applications; however, it is <125 ms for some critical medical applications. Similarly, sensor nodes should be capable of reliable data transmission, and the collected information must be sent to the coordinator node with a defined probability of success because lost or missing data could affect data processing and relevant decision-making. However, there are trade-offs among reliability, delay and energy consumption. If we want higher reliability, then delay and power consumption will increase. Moreover, less delay and high reliability tend to consume more energy.

Table 1 shows values of the required data rate and latency for wearable sensors for patient monitoring systems.

Meanwhile, several survey papers containing various aspects of WBASNs have been published [35,36]. In [1], the authors provided a detailed survey of WBASNs by considering the IEEE 802.15.6 standard. It covers all the aspects, including applications, standardisation efforts and physical, MAC and routing protocols. The authors discussed the state of the art of WBASNs in terms of application requirements and then linked it to the routing protocols. A detailed comparison of routing protocols is provided by considering delay, reliability, throughput and energy consumption. Further, open issues in this area are highlighted. The authors [ focused on the energy and power issues of WBASNs. Various protocols are discussed and summarised by considering power and energy consumption as key performance contributors [37,38,39,40,41,42].

In comparison to the existing surveys in the WBASN area, the summary of contributions for this paper is presented as follows:We conduct a comprehensive survey of MAC and routing protocols of WBASNs by considering the patient monitoring systems under the standards IEEE 802.15.4 and IEEE 802.5.6; in contrast, most of the published surveys of WBASNs only focused on IEEE 802.15.6. The reason for selecting IEEE 802.15.4 along with IEEE 802.15.6 is that most industrial implementations use IEEE 802.15.4 for WBASNs. The IEEE 802.15.6 standard as a ready solution is still not available.The categorisation of MAC protocols for WBASNs is provided based on the literature from the period 2005 to 2019 for the IEEE 802.15.4 and IEEE 802.15.6 standards. Based on the provided categorisation, a comparative analysis of the MAC protocols is provided; these protocols optimise the IEEE 802.15.4 and IEEE 802.15.6 standards in terms of delay reliability, throughput, mobility, interference and energy consumption. In contrast, the published surveys of WBASNs cover one or two categorisations of MAC protocols by only considering IEEE 802.15.6, which is still not widely available, and most patient monitoring systems use IEEE 802.15.4.We provide a categorisation of the routing protocols for WBASNs for the standards IEEE 802.15.4 and IEEE 802.15.6 from the period 2005 to 2019. Although similar categorisation can be seen in the published surveys, in the published surveys, the discussion regarding open issues and challenges for each category is missing. We provide a comparative analysis of the routing protocols under each categorisation by considering various performance metrics, including delay, reliability, throughput and energy consumption. Further, under each categorisation, we provide open issues and challenges.We provide a detailed background of WBASNs, including architecture, topologies, standards, application requirements for chronic diseases, the benefits and use of various frequency bands, comparative analysis of WBASN’s available technologies, including LoRa and NB-IoTs, etc.

Therefore, the essence of such a multi-aspect survey in WBASNs is beneficial to the research community as it provides recent trends on WBASNs. Figure 2 provides the taxonomy of this survey.

The rest of the paper is organised as follows: Section 2 presents the background of WBASNs, including its architecture, requirements and existing technologies and standards. Section 3 introduces the proposed taxonomy of MAC protocols with categorisation and comparative analysis in terms of delay, reliability, throughput, mobility, interference and energy. Section 4 provides the categorisation of routing protocols, the comparative analysis of various recent protocols under each category. Section 5 provides a discussion about open issues and challenges. Section 6 concludes the survey paper.

## 2. Background

WBASNs are considered the sub-field of wireless sensor networks (WSNs). In 1995, Zimmerman introduced the concept of WBASNs, where physiological information is exchanged among devices placed inside or near the human body. He also suggested using WPANs for physiological data collection from devices. In their deployed testbed, communication channel properties, the establishment of reliable links and the network connection of devices to the application were done at the physical layer, the data link layer and the network layer, respectively. Low carrier frequency is used to minimise energy consumption. Figure 3 explains a setup that consists of a WPAN transmitter and receiver and is connected to the human body. The current flows with the help of a biological conductor, and to prevent shorting, “earth ground” is used. The “earth ground” is considered an important aspect of WPAN devices, and it is suggested that the best location for these devices is near the feet.

Various communication protocols and mechanisms are developed for WSNs, but their use in WBASNs is not suitable due to the different environments of the human body. There are a few common features between them, which include network structure, energy efficiency and multi-hop communications [43,44,45,46]. The comparison between WBASNs and WSNs is made based on five aspects, i.e., node features, network size, limited resources, accessibility and mobility [47,48].

### 2.1. Comparison between WSNs and WBASNs

(1)Node Identification

Node identification refers to the process of assigning a unique identification ID to a node. The IDs have local significance in WSNs and WBASNs. It is expected that the number of deployed nodes in WBASNs is less than in WSNs, where hundreds of nodes are deployed. Hence, fewer numbers of bits are used to identify the WBASN nodes. A fewer number of bits as a node identification reduces the processing time and energy consumed.

(2)Node Size

In WBASNs, smaller-size sensor nodes are used, whereas, in WSNs, large-size nodes can be used according to the requirement of the scenario. In WBASNs, two types of sensor nodes are used, i.e., implanted and wearable. The size of implanted nodes is very small. Figure 4 shows different implanted and skin sensor nodes. Their purpose is to read text and convert the words into voice signals.

(3)Network Size

Usually, a WBASN consists of a limited number of nodes, which vary between 6 to 12, whereas a WSN is made of hundreds of nodes. In WBASNs, the transmission range is selected according to the height of the human body (up to a few meters), and all the sensor nodes send data to a BAN coordinator (BANC) that transmits the data to the destination. In WSNs, the transmission range can be 100 m, and a dedicated node cluster head is used as the coordinator. Overall, the network area of WBASNs is several meters due to its low transmission range, and it requires low transmission power, which is not harmful to body tissues. In WSNs, the network area is hundreds of meters due to its high transmission range and requires high power for transmission.

(4)Limited Resources

In WBASNs, the size of sensor nodes is tiny, and they have limited resources in terms of bandwidth, energy source, processing speed and memory; in contrast, in WSNs, due to the bigger size of the nodes, the resources are not as limited as in WBASNs.

(5)Mobility

As WBASNs are deployed on the human body, internal and external body mobility creates complexity. Hence, the protocols need to support mobility in WBASNs. In WSNs, the network structure is usually static.

### 2.2. WBASN Components

The WBASN’s node structure contains various modules, including an energy source, a processor, memory, a transceiver, a sensor, actuators and an operating system [50,51,52]. Figure 5 shows the structure of the WBASN node.

(1)Energy Source

In WBASNs, a small battery size limits the energy source and allows very low-power levels in order to increase the lifetime of the sensor node.

(2)Processor

The function of the processor is to manage all the computation activities. Various companies make microcontrollers for WBASNs. In this context, MSP430 from Texas Instrument (TI) is an example; it is considered the world’s most popular ultra-low-power microcontroller with a 16-bit microcontroller platform. The speed of this processor varies from 8 to 15 MHz.

(3)Memory

Memory capacities vary in WBASNs, e.g., a node with MSP430 contains up to 64 KB RAM with up to a maximum of 512 KB flash memory.

(4)Transceiver

A transceiver is a component that can transmit and receive data. Usually, WBASNs consist of a CC2420 chip that is useful for low-power data communications.

(5)Sensors

Generally, the sensing module contains various sensor nodes that are used to monitor the physiological parameters of the human body.

(6)Actuators

Actuators take action against the data received from the sensors, e.g., upon receiving some critical data regarding diabetes, the actuators can inject insulin.

(7)Operating System

TinyOS is a popular open-source operating system used in WBASNs. As TinyOS’s design supports low-power communications, it is suitable for WBASN devices.

### 2.3. WBASN Topologies

The network topologies describe the data communication structure among the network’s nodes. In WBASNs, different types of topologies are used, including peer-to-peer (P2P), mesh, cluster tree, hybrid and star. According to the requirement of applications, a topology is selected. These requirements include scalability, robustness, energy, reliability, latency and mobility. According to the IEEE 802.15.4 standard, functionality-wise sensor nodes are divided into two types, i.e., full-function sensor nodes (FFSNs) and reduced-function sensor nodes (RFSNs). The FFSNs are capable of routing functions, whereas RFSNs can only do peer-to-peer communication. Usually, RFSNs are deployed when energy is a critical issue. The advantages and disadvantages of these topologies are discussed in [53,54].

### 2.4. WBASN Requirements

WBASNs face various challenges due to the diverse nature of applications, and these requirements are different from other wireless network technologies. Table 2 describes these characteristics and requirements [55,56].

### 2.5. WBASN in Healthcare

Healthcare is facing a challenge as remote monitoring will be needed for a population of 761 million people in 2025 [56]. Additionally, the number of patients with chronic diseases has increased, so there is a need to provide quality of life in terms of healthcare. Patient data monitoring is one of the most important applications in healthcare. Further, continuous monitoring of patients in indoor and outdoor environments proves to be very useful for doctors to extract useful information for treatment and care. Hence, WBASNs are used for remote healthcare and monitoring in various environments such as hospitals, ambulatory, emergency and elderly care centres, etc.

For a chronic disease patient, the formal procedure of routine visits is required to monitor the progress, development of complications or relapse of the disease. Questions such as what, how and when to monitor are crucial for treatment. In this context, various biosensors are used for monitoring the patient’s physiological conditions in order to receive relevant information regularly. Table 3 provides examples of diseases with measurable physiological parameters and usable sensor types.

### 2.6. WBASN Global Connectivity

WBASNs are capable of interaction with the internet and other existing wireless technologies, including ZigBee, Bluetooth, WLANs and cellular networks, etc. There are various ways to connect WBASNs to the internet; usually, it connects with the help of ambient sensors. Figure 6 shows a generic scenario of WBASN global connectivity.

The IPv6 over low-power wireless personal area networks (6LoWPAN) standard helps to connect the devices to the internet. The concept of 6LoWPAN follows the idea that internet protocol can be applied even to small, low-power devices that have limited processing capabilities to make them a part of the IoTs. 6LoWPAN is a working group of the Internet Engineering Task Force (IETF) and defines encapsulation and header compression mechanisms that allow IPv6 packets to interact with IEEE 802.15.4 standard. The focus of IP networking for low-power radio communication is those applications that require wireless internet connectivity at lower data rates. The Thread consortium is a consortium that makes the protocol that is run in the 6LoWPAN standard.

### 2.7. WBASN Standards

The implementation of WBASNs is usually done using WPAN communication protocols, including ZigBee (IEEE 802.15.4), IEEE 802.15.6 and Bluetooth, etc. Various other wireless technologies can be potentially used for WBASNs. Standardised bodies such as the IEEE, the International Society of Automation (ISA) and the IETF have created new wireless standards, and most of these standards are available as commercial products. In the literature, there are various contributions towards lower-power network-based devices for WBASNs, such as ZigBee, 6loWPAN [57], 6lo [58], 6tisch [59], ISA SP-100 [60], IEEE 802.15.4 [61] and IEEE 802.15.6 [62].

(1)IEEE 802.15.6

IEEE 802.15.6 was developed by a task group in 2007 for the standardisation of WBANs and approved in 2012 [63,64,65]. This standard is used for implantable as well as wearable sensors and works at lower frequencies within a short range. This standard presents a MAC and physical layer design to support various applications, including medical and non-medical applications. Medical applications refer to a collection of vital information in real-time (monitoring) for the diagnoses and treatment of various diseases with the help of different sensors (accelerometer, temperature, BP and EMG, etc.). It defines a MAC layer that works with three PHY layers, i.e., human body communication (HBC), ultra-wideband (UWB) and narrowband (NB). IEEE 802.15.6 also provides a specification for the MAC layer to access the channel. The coordinator divides the channel into superframe time structures to allocate resources. Superframes are bounded by equal-length beacons through the coordinator. Table 4 describes the different frequency bands for IEEE 802.15.6.

(2)IEEE 802.15.4

IEEE 802.15.4 is the standard that states the physical layer and MAC layer functionality for LR-WPANs. It was established by the IEEE 802.15 working group, which provided the basis for the WPAN standard. IEEE 802.15.4 discusses different device roles, including full-function devices and reduced-function devices. The physical layer of IEEE 802.15.4 uses three frequency bands, including 2.4 to 2.4835 GHz with 16 channels, 902 to 928 MHz with 10 channels and 868 to 868.6 MHz with a single channel. The data link layer is made of two sub-layers, MAC and logical link control. The MAC layer manages activities such as beacon management, GTS management, and channel access.

The standard acts as a basis for ZigBee, WirelessHART and ISA 100.11a, etc., and uses 6LoWPAN to provide connectivity with the internet. Various WBASN standard-based commercial products operate with the IEEE 802.15.4 standard. The standard claims to provide energy-efficient communication and provides appropriate throughput, limited latency and acceptable reliability. A survey in 2016 of 100 users found that IEEE 802.15.4 was used by 63% of them for a variety of IoT applications, which indicates its market acceptability and strong relation with IoT applications [66]. IEEE802.15.4 with ZigBee has become a popular industrial choice for IoT applications. Moreover, various task groups of IETF, e.g., “routing over low power and lossy networks”, have recommended using IEEE 802.15.4. The task groups of IEEE 802.15.4 can be summarised as follows.

IEEE 802.15.4a provides an amendment to IEEE 8021.5.4 by incorporating two PHYs, including UWB and Chirp. These PHYs provide features such as good throughput, power efficiency and different data rates. The IEEE 802.15.4e standard provides an extension to the MAC layer functionality of IEEE 802.15.4 to make it more capable for industrial, commercial and medical applications by using the concept of multi-channels.

(3)ZigBee

The ZigBee standard operates under the umbrella of the ZigBee Alliance, which consists of more than seventy members from the communication industry. ZigBee is a wireless mesh network standard with the characteristics of low power, low cost, energy efficiency and limited latency, which makes a strong case for its use in industrial and medical applications. ZigBee chips are integrated with microcontrollers, and they operate under various ISM frequencies bands, including 2.4 GHz, 784 MHz (China), 868 MHz (Europe) and 915 MHz (the USA and Australia), with data rates from 20 to 250 Kbps.

ZigBee works on the network layer by supporting star, tree, and mesh topologies. It works under the same guidelines of IEEE 802.15.4, i.e., a central coordinator as a controlling entity. ZigBee is developed over the physical layer and MAC mechanisms of the IEEE standard 802.15.4 and adds four additional key components, i.e., the network layer, the application layer, manufacturer-defined application objects and ZigBee device objects. ON World, a famous technology research firm, published a report that ZigBee’s share of IEEE 802.15.4-based IoT applications is growing day by day and that, by 2020, ZigBee will be used in up to 80% of IEEE 802.15.4-based IoT devices [67]. Table 5 provides a comparative analysis of available wireless technologies for WBASNs.

### 2.8. Power Consumption

The miniatured batteries can be used for WBASN nodes. For efficient energy utilisation, new energy-efficient communication protocols are used at the MAC and network layers. These protocols reduce power consumption by introducing duty-cycle mechanisms. Table 6 describes the power consumption, data rate and battery lifetime comparison of WBASNs with other existing wireless technologies. The data rate and communication protocol’s carrier frequency have a huge impact on power requirements/consumption, and generally, higher frequency + higher data rate will mean higher power consumption.

## 3. Review of WBASN MAC Protocols for IEEE 802.15.4 and IEEE 802.15.6

The interest in WBASNs for remote patient monitoring has increased considerably. It is evidenced that the medium access method used in WBASNs plays a vital role in fulfilling the specific set of QoS requirements for biomedical devices. These QoS requirements are a specified set of time-bound data transmission, data rates, reliability and energy consumption, etc. In the last few years, many MAC protocols for WBASNs have been proposed for medical applications. The WBASN MAC protocols are classified into three broader categories based on channel access mechanisms, i.e., contention, scheduled and hybrid access mechanisms.

MAC protocols play a critical part in providing QoS for WBASNs by extending network lifetimes, avoiding packet collisions, reducing overhearing and idle listening, etc. Generally, WBASN MAC protocol characteristics include energy efficiency, scalability, low latency, fairness in terms of channel access, throughput and jitter, etc. Based on channel access mechanisms, the MAC protocols are categorised into three types, i.e., schedule, contention and hybrid-based access mechanisms. For WBASNs, mostly hybrid-based mechanisms are used, due to their flexibility, to adjust light and heavy data traffic. The hybrid MAC protocols combine the benefits of both access mechanisms, i.e., contention-based MAC and schedule-based MAC. The IEEE 802.15.4 and IEEE 802.15.6 MAC protocols have evolved from the idea of an adaptive MAC protocol that combines slotted ALOHA and TDMA, presented in the 1970s [68,69], to attain maximum throughput. Therefore, several optimisation mechanisms for the IEEE 802.15.4/IEEE 802.15.6 MAC protocols have been proposed.

The existing literature regarding the optimisation of the IEEE 802.15.4/ IEEE 802.15.6 MAC protocols is categorised in Figure 7.

### 3.1. MAC-Layer-Based and Parameter-Tuning-Based Approaches

These approaches recommend that minimum modification should be made to the standard so that appropriate benefits can be achieved from the strengths of the standard. In this context, according to the requirements of the application, parameters should be tuned properly. The performance of the slotted CSMA/CA mainly depends on four parameters, namely, the minimum backoff exponent (macMinBE), the maximum backoff exponent (macMaxBE), the contention window (CW) value and the maximum number of backoffs (macMaxCSMAbackoffs). The advantage of this approach is that it does not require a complex modification of the existing protocol. The drawback of this approach is that this optimisation may only be specific to some applications. The details of such optimisations, based on parameter tuning, can be found in [70,71,72,73,74,75,76,77,78,79,80,81,82,83]. Overall, the higher the number of retransmissions, the better the reliability of the data transmission. Retransmission occurs when a transmitting device does not receive an acknowledgement from the receiver. There are different reasons for not receiving the acknowledgement, i.e., data packet loss due to a collision on receiving side, acknowledgement loss and late reception of the acknowledgement, etc. A high number of retransmissions assures reliability, but, at the same time, it causes a delay in network performance as retransmissions will involve channel access to the same packet; it also requires bandwidth, which will affect the performance of other nodes. The traditional profile of IEEE 802.15.4 and IEEE 802.15.6 imposes restrictions by giving default values for these MAC layer parameters.

The discussion above concludes that IEEE 802.15.4/IEEE 802.15.6 standards are capable of supporting time-critical applications by tuning the MAC parameters to more suitable values. There are few performance trade-offs between reliability and latency and, similarly, between delay and energy consumption. Hence, it can be observed that these MAC parameters have a significance impact on network performance. There is a need to understand the appropriate setting of these parameters to optimise the performance of biomedical applications that require specific data rates and latencies.

### 3.2. Cross-Layer-Based Approaches

These approaches use information from other layers to tune the MAC layer parameters. Although such approaches fulfil the requirement of the applications as they depend on the information of the other layers, there is a high chance that delay will occur, which is not tolerated for medical applications.

Adaptive access parameter tuning (ADAPT) [84] is proposed as a cross-layer protocol to attain energy efficiency and reliability. The idea of ADAPT is to initially understand the application’s reliability levels and then perform the parameter tuning. ADAPT operates under an adaption module that is capable of interacting with other layers of the ZigBee stack. This adaption module collects information from different layers to optimise performance. ADAPT perceives requirements from the application layer and maps them to the MAC layer so that suitable tuning can be performed to the parameters, including macMinBE, macMaxCSMABackoffs and MacMaxFrameRetries. It considers single-hop and multi-hop scenarios. An optimisation problem is developed with the objective function of minimising energy. The proposed model works on a few constraints, i.e., the delay value should not increase more than the threshold and bandwidth allocation to the nodes in the network. An adaptive-variable scheme that considers the requirement of the application and the PHY layer is used to allocate the bandwidth. The framework jointly optimises the performance in terms of energy and bandwidth.

Timely, reliable, energy-efficient and dynamic (TREnD) [85] is a cross-layer protocol that focuses on various industrial applications. TREnD enables interaction among the routing algorithm, the MAC layer and power modules to achieve the required reliability and latency. TREnD operates in inter-cluster and intra-cluster architecture by accommodating local routing and dynamic routing. A hybrid MAC protocol with TDMA/CSMA is used with local routing cases. The nodes wake up within their predefined slot time in a TDMA-based approach to save energy. The TDMA mechanism is applied in such a way that different clusters are synchronised with it. The receiving nodes send the multicast beacons to stay in the topology.

### 3.3. Duty-Cycle-Based Approaches

Various research studies have optimised the performance of the slotted mode of IEEE 802.15.4 in terms of latency and energy consumption by adjusting the duty cycle mechanism accordingly. These approaches deal with channel access management during the active and inactive periods of the superframe to make power utilisation efficient. The advantage of this approach is to increase the network lifetime by managing the duty cycle period.

AMPE fine-tunes the duty cycle by considering the occupancy rate of the superframe order (SO). The dynamic superframe adjustment algorithm [86] adjusts the duty cycle using two parameters, including the superframe occupancy rate of the SO and the collision rate. Duty cycle adoption [87] utilises the buffer occupancy and queuing delay statistics to optimise the duty cycle mechanism. The adaptive algorithm optimises the dynamics [88] and adjusts the superframe’s active periods using the comparison number of received packets in different superframes. The duty-cycle learning algorithm [89] improves the duty-cycle mechanism based on the offered traffic load.

### 3.4. Priority-Based Approaches

These approaches improve IEEE 802.15.4/ IEEE 802.15.6 MAC by considering the priorities of the channel access mechanism. These approaches introduce mechanisms to recognise the priority of the nodes. The weighted-fair-queue (FQ-CSMA/CA) [90] algorithm differentiates between the alarm signals in an emergency event and normal signals. The preference for the signals is given based on urgency. The algorithm aims to reduce the latency for alarm signals without disturbing the traffic of normal signals by introducing weighted queues. Overall signals are divided into five categories according to their urgency, including sensor data traffic, ACK traffic, command control traffic, system settings and alarm signal traffic.

A Markov-based analytical model that deals with nodes having a different set of access parameters, with two priority classes, including high and low priorities, has been designed [91] for CAP. These priorities consider contention widow values to adjust the prioritised traffic, which means a prioritised node will get more chance to access the channel.

Differentiated services are delivered to prioritised traffic classes, which are made for critical data traffic. The work in [92] tunes the MAC layer transmission parameters, including macMinBE, macMaxBE and the initial size of CW. This tuning is applied based on priority; mostly, data traffic is assigned the lowest priority, whereas alarm reports, command frames and GTS data are considered high priority. Lower back-offtime-period values are given to high-priority traffic and vice versa; moreover, for high-priority traffic, a priority queue is introduced.

An explicit priority scheme has been designed [93] for IEEE 802.15.4 by categorising the transmitting nodes into critical and non-critical nodes. Critical nodes have urgent data to send, whereas the non-critical nodes have normal traffic data and can tolerate the delay. A secondary beacon is used by the nodes to inform the coordinator about the urgency, and the coordinator only allows those nodes in CAP that have informed and requested it to send urgent data. Moreover, after receiving the secondary beacon, the coordinator generates the primary beacon and informs the other nodes about activities in the upcoming CAP.

Two mechanisms have been proposed to provide differentiation services to IEEE 802.15.4-based nodes in saturation conditions, including contention window differentiation (CWD) and backoff exponent differentiation (BED). Nodes are categorised into various priority classes, i.e., emergency and high bandwidths, etc. These priorities are assigned based on data traffic. Overall, CW and binary exponent/backoff values are used to provide these prioritised services. Further, CWD and BED are used to make another scheme known as the backoff counter selection (BCS). BCS is used to select the next backoff period when it is medium-busy. A Markov-based model is developed to validate the performance of CWD and BED for IEEE 802.15.4.

### 3.5. Superframe Modification Approaches

These approaches improve the slotted CSMA/CA protocol of IEEE 802.15.4/ IEEE 802.15.6 by proposing modifications in the superframe structure.

An emergency beacon service is used to manage the normal, emergency and periodic data transmission services in CAP through a contention access mechanism [94]. The coordinator is responsible for the transmission of the emergency beacon to handle the emergency services in the CAP period by modifying the superframe structure. The energy consumption analysis shows that the existing superframe structure is not enough to manage emergency traffic. The priority-based load adaptive MAC (PLA-MAC) protocol [95] offers four data classes, including ordinary, delay, reliability and critical data packets. Few dedicated slots are introduced in the superframe to handle emergency data traffic. Low-delay traffic-adaptive medium access control (LTD-MAC) [96] extends the contention periods in CFP. However, this protocol stops the transmission after all channels are occupied, which causes data loss. Adaptive and real-time GTS allocation (ART-GAS) [97] handles the differentiated services by adaptively introducing GTS slots in the superframe duration. It is noticed that the performance of other nodes with normal data traffic suffers by frequently using these GTS slots. Differentiated service classes have been proposed, including emergency-TDMA (ETDMA), medical contention access periods (MCAP) and normal-TDMA (NTDMA). This scheme works well; however, managing slot assignment for different nodes that desire to access the channel is complex and energy-consuming. Fuzzy control medium access (FCMA) [98] decides on slot assignment in CAP and CFP on the bases of fuzzy rules, which are made on priority and data rates. It works in three phases, including data acquisition, fuzzy-logic control and implementation. Priority-based adaptive timeslot allocation (PTA) [99] divides the CAP channel into chunks. Different slots are assigned to the nodes according to the priorities; however, the mechanism is expensive in the context of limited latency and energy consumption.

To conclude the above discussion, there are different optimisation mechanisms to improve the IEEE 802.15.4/ IEEE 802.15.6 protocol. Table 7 summarises the categorisation of optimisation approaches with their possible advantages and disadvantages.

Table 8 presents and summarises some of the proposed protocols based on the categorisation presented in Figure 6. Moreover, we have shown indications as delay (D), reliability I, energy efficiency I), throughput (T), collision handling (C), priority (P), mobility (M), interference (I) and scalability (S).

## 4. Review of the Routing Protocols

The IEEE 802.15.4 standard defines the operational functionalities of LR-WPANs [124]. As it supports peer-to-peer and star topologies, it is considered a popular standard for low-power sensor monitoring applications for healthcare, factories and disaster areas, etc. [125,126,127,128].

A detailed comparison analysis regarding hop selection is provided in [129]; it concludes that multi-hop communication leads towards better network performance in terms of energy consumption, with a minor decrement in the packet delivery ratio; this can be ignored because less energy consumption will increase the network lifetime in WBASNs. The discussion in [130] indicates that a route using several short hops provides better energy consumption and network lifetime than routes with long hops. The research discussed various reasons to prove that routing strategies with the involvement of long hops are more successful in terms of power efficiency. For single-hop access communication, it was concluded that the node with more distance from the base station consumes high energy and, ultimately, less network lifetime. The work in [131] shows that single-hop communication may not provide reliable communication for WBASNs, whereas, for multi-hop communication, they define separate routes with minimum delays and high overall. WBASN routing protocols have many challenges, including limited resources, energy efficiency, time-bounded delay (<50–250 ms for medical applications), traffic priorities and the unreliability factor of low-power wireless links. The routing protocols for WBASNs can be categorised into four broader categories, as shown in Figure 8.

Multi-hop routing protocols for WBASNs are mostly classified into four categories: QoS-based, cluster-based, cross-layer-based and link-quality-based. The heterogeneous nature of sensors requires prioritised scheduling and forwarding mechanisms, which are provided by QoS-based routing protocols through QoS priority routing QoS modules for delay and reliability. Although these protocols provide QoS, their designs are complex, which makes them less scalable. Cluster-based protocols are designed to reduce energy consumption for sensor nodes where, in a limited region, a cluster is created, and a cluster head is selected for communication outside the cluster. This increases the overall network lifetime by reducing the multiple communication attempts of various nodes through cluster heads. However, the overheads involved in the cluster-head selection among nodes make it less energy efficient and, ultimately, less scalable; if a large number of nodes make a network, then the head selection process will get complex and consume more energy. In cross-layer approaches, the network layer involves different layers to perform energy-efficient and delay-bounded routing; however, due to dynamic network conditions and mobility, these protocols do not deliver the expected results.

### 4.1. QoS-Based Routing Protocol Comparison

Table 9 shows a comparison between QoS routing protocols. The comparison parameters include QoS, mobility, scalability, energy efficiency and the used methodologies.

The aforementioned QoS-based routing protocols show the following aspects:WBASNs require prioritised QoS mechanisms at the network layer to handle the heterogeneous nature of various body sensors.Geographical position and residual energy are the most important metrics for next-hop selection.End-to-end delay, reliability and packet delivery ratios are the most considered network performance parameters.

### 4.2. Cross-Layer-Based Routing Protocol Comparison

Cross-layer routing protocols combine the challenges of the network layer with other layers. Even though these protocols have low energy consumption, high throughput and fixed end-to-end delay, they cannot supply high performance in scenarios with high path loss and body motion. Table 10 provides a comparison among cross-layer routing protocols; the comparison parameters include congestion avoidance, mobility, scalability, energy efficiency and the used methodologies.

The aforementioned cross-layer routing protocols show the following aspects:Energy consumption, end-to-end delay and throughput are the main considerations.Most of them agree to a tree-based approach to improve energy consumption.Time division mechanisms are also used to provide channel guarantees.Transmission power should be adopted according to the distance.

### 4.3. Cluster-Layer-Based Routing Protocol Comparison

Cluster-based routing algorithms are those that divide nodes in WBANs into different clusters and assign a cluster head for each cluster. Data are routed through the cluster heads from the sensors to the sink. This class of routing protocols aims to decrease the number of direct transmissions from the sensors to the base station. However, the huge overhead and delay relative to cluster selection are the main drawbacks of these protocols. Table 11 shows a comparison between cluster-based routing protocols; the comparison parameters include congestion avoidance, mobility, scalability, energy efficiency and the used methodologies.

The aforementioned cluster-based routing protocols highlight the following aspects:Most of them are scalable.Efficient algorithms are used for cluster-head selection and for optimising end-to-end path selection.

### 4.4. Link Quality-Based Routing Protocols Comparison

The link quality-based routing protocols are robust and select the next hop based on link quality information such as energy, hops, processing and memory. Additionally, low-power radios are very sensitive to noise, interference and multipath distortions. For the low-power WBASNs, link quality is considered a crucial metric for the selection of the next hop [152,153,154].

There are many link-aware routing protocols of WBASNs that can be adopted with IoTs [155,156,157,158,159,160,161]. These protocols mostly use IEEE 802.15.4 or IEEE 802.15.6 as MAC and PHY layer standards. Various methods are used to compute the link quality for these routing protocols [162]; however, RSSI and link quality indicator (LQI) are considered strong candidates, as recommended by the ZigBee standard [163], IETF 6LoWPAN WG [130] and IETF ROLL WG [139], etc.

Two ways are used to measure the link quality, including packet-based techniques and radio-hardware-based techniques [164]. Packet-based techniques compute the link quality using a number of received and estimated packets in a specified time [165]. Control packets are used to implement this approach [166]. As the packet-based technique maintains the state information, it needs more processing time, memory and energy [167].

The routing protocol for low-power and lossy networks (RPL) is a popular routing protocol used for IoT-based WBASNs, and it became standard in 2011 by IETF. RPL works with IPv6 to make a complete IoT-based network. At the MAC and PHY layers, RPL uses IEEE 802.15.4. Node status information is used by RPL to decide on the next hop. The node status information includes residual energy, memory and link quality. RPL is recommended for healthcare applications by IETF and ZigBee [168].

For WBASNs/IoTs, the proposed routing protocols have mostly evolved through ad hoc on-demand distance vectors [169]. The link-aware category depends on the link state information for routing. These protocols use parameters such as end-to-end delay, RSSI, PRR and power, which affect the performance of data delivery.

Routing by energy and link quality (REL) is an IoT-based routing protocol that was developed for use with WSNs for patient monitoring systems. The REL considers issues such as energy utilisation, reliability and latency. These low-power radios are sensitive to the interference generated by nearby devices. REL uses the link quality metrics, i.e., LQI and RSSI, to overcome the sensitivity issues; this is helpful in providing reliability. In REL, the next hop is selected based on a set of matrices, including link quality, residual energy, hop count and load balancing.

The link-quality-based lexical routing (LABILE) protocol proposes a routing algorithm that uses the lexical structure with link quality information [170]. LAIBLE uses the LQI value to provide link reliability. LABILE categorises the computed value of link quality as “good” or “bad” according to predefined threshold values. The LABILE protocol ignores the energy efficiency metric while selecting the next hop.

## 5. Challenges and Open Issues

The main challenges and open issues noticed from the performed analysis of WBASN MAC and routing literature are as follows.

### 5.1. Challenges/Open Issues for MAC protocols

In MAC protocols, the energy waste is due to idle listening, collisions, packet overheads and overhearing useless traffic. As a WBASN may consist of multiple physiological sensors that require different data rates, the selection of the appropriate radio frequency (RF) is a crucial part of deploying patient monitoring systems. The heterogeneous nature of biomedical sensors in terms of sensing and transmitting data make the required QoS more complex for the MAC layer as it may need to send some data with high priority, such as ECG data in emergency scenarios. Hence, a priority mechanism should be adopted according to the nodes’ requirements rather than providing services based on predefined values. A balanced MAC protocol is required for WBASNs that can provide QoS (bounded delay, required throughput, minimum energy consumption and minimum collision) parameters simultaneously.

### 5.2. Challenges/Open issues for Routing Protocols

QoS-based routing protocols show the following shortcomings/open issues:

Scalability is the main issue for these protocols. Routing updates and maintenance mechanisms create overheads due to inappropriate dissemination and “hello” message sizes. The timing of route updates is ignored, which ultimately affects mobility support. Most QoS-based routing protocols ignore the congestion control and avoidance mechanisms in conjunction with prioritised QoS. Route failure cases are ignored, and packets are dropped in the case where a node does not provide the required QoS.

Cross-layer routing protocols show the following shortcomings/open issues:

The heterogeneous nature of body sensors demands prioritised QoS, which is not considered in these protocols. Scalability is a challenge for these protocols as their performance decreases in dense scenarios. Routing updates and maintenance mechanisms create overheads due to inappropriate dissemination. Mobility effects are ignored, whereas most of the WBASN applications demand mobility. Link reliability consideration is ignored.

Cluster-based routing protocols show the following shortcomings/open issues:

The heterogeneous nature of body sensors demands prioritised QoS, which is not considered in these protocols and, hence, not appropriate for most WBASN scenarios. In large clusters, the cluster-head selection algorithm causes energy overheads.

Link-quality-based routing protocols show the following shortcomings/open issues:

Link quality is usually measured as a single value, such as a received signal strength indicator (RSSI) or link quality indicator (LQI). However, LQI/RSSI only represents a snapshot at a specific point in time for one link between two nodes and lacks any additional information about the remaining energy, hop count and end-to-end information. Thus, there is still an urgent need to find a reliable scheme to estimate the end-to-end link quality based on information from different layers.

## 6. Conclusions

This survey provides a detailed review of recent research activities on WBASNs in the context of MAC and routing protocols. Critical challenges and potential future work for the MAC and routing protocols are identified. Although extensive research on WBASN communication has been provided, there are various pressing issues to be solved in the future. Most of these issues are driven by the applications, as each application has its specific set of requirements for communication. Some of the applications do not require a high data rate; however, they are more sensitive towards delay, and vice versa. To achieve the stringent QoS requirements, MAC and routing protocols can play a key role. The MAC and routing protocols play their role by fulfilling the stringent QoS requirements for remote patient monitoring applications. It is evidenced that the medium access method used in WBASNs plays a vital role in fulfilling the specific set of QoS requirements for biomedical devices. There are different optimisation mechanisms to improve the IEEE 802.15.4/ IEEE 802.15.6 protocol, including parameter-tuning, duty-cycled, prioritised-based, superframe optimisation and cross-layer mechanisms. However, a QoS performance trade-off exists among these optimisation mechanisms; the details are provided in Table 6 and Table 7. Routing protocols for WBASNs are mostly classified into four categories: QoS-based, cluster-based, cross-layer-based and link-quality-based. The heterogeneous nature of sensors requires prioritised scheduling and forwarding mechanisms that are provided by QoS-based routing protocols through QoS priority routing and QoS modules for delay and reliability. Although these protocols provide QoS, their designs are complex, which makes them less scalable. Cluster-based protocols are designed to reduce energy consumption for sensor nodes where, in a limited region, a cluster is created, and a cluster head is selected for communication outside the cluster. This increases the overall network lifetime by reducing the multiple communication attempts of various nodes through cluster heads. However, the overheads involved in the cluster-head selection among nodes make it less energy efficient and, ultimately, less scalable; if a large number of nodes make a network, then the head selection process will be complex and consume more energy. In cross-layer approaches, the network layer involves different layers when performing energy-efficient and delay-bounded routing; however, due to dynamic network conditions and mobility, these protocols do not deliver the expected results. In conclusion, progressive research for this essential technological area has a crucial role in future well-being; therefore, this detailed survey will act as a source of inspiration for future developments in WBASNs.

## Figures and Tables

**Figure 1 sensors-22-08279-f001:**
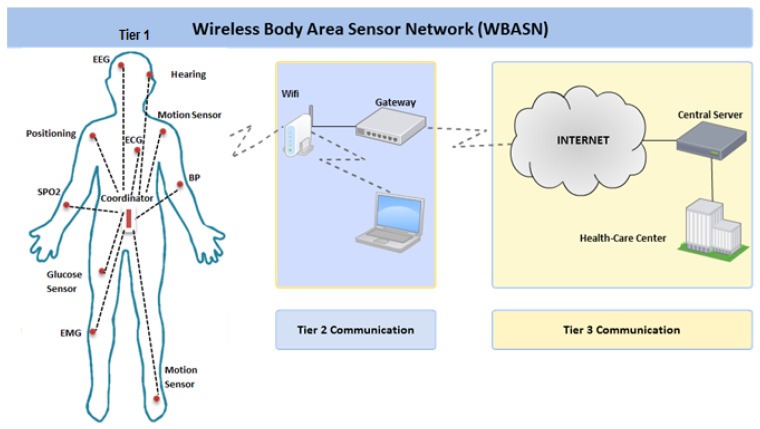
Wireless body area sensor network overview.

**Figure 2 sensors-22-08279-f002:**
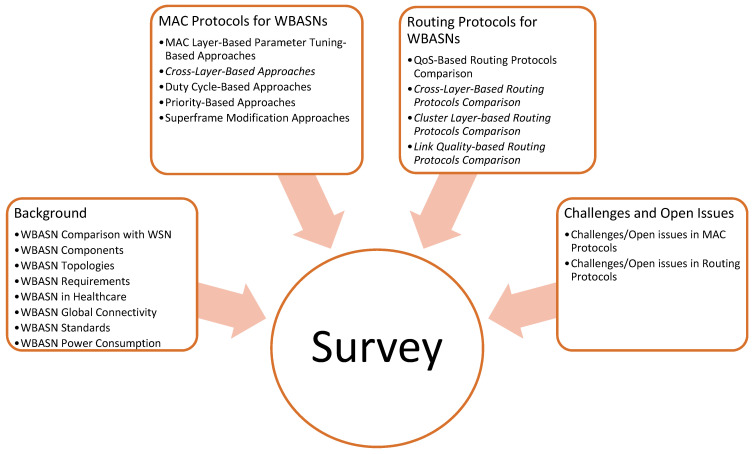
Taxonomy of this survey.

**Figure 3 sensors-22-08279-f003:**
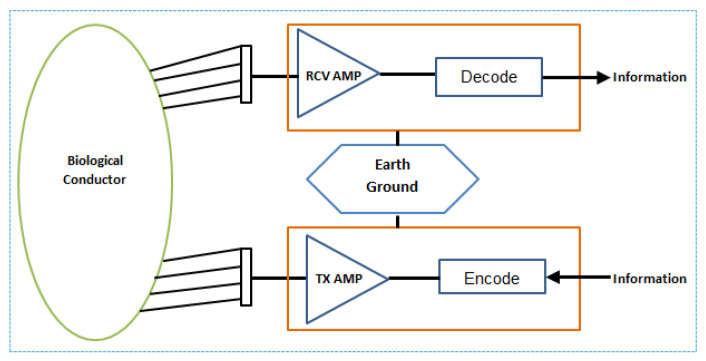
Block diagram of the PAN system.

**Figure 4 sensors-22-08279-f004:**
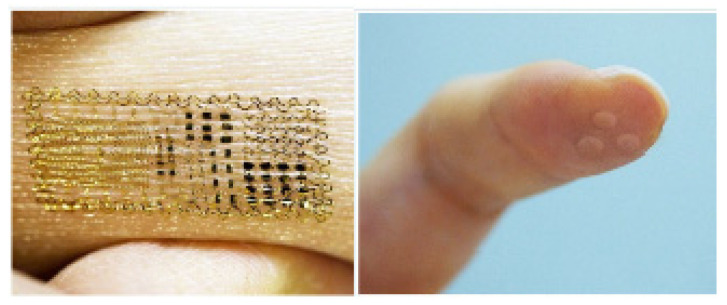
Smart skin sensors and finger-implanted sensors [49].

**Figure 5 sensors-22-08279-f005:**
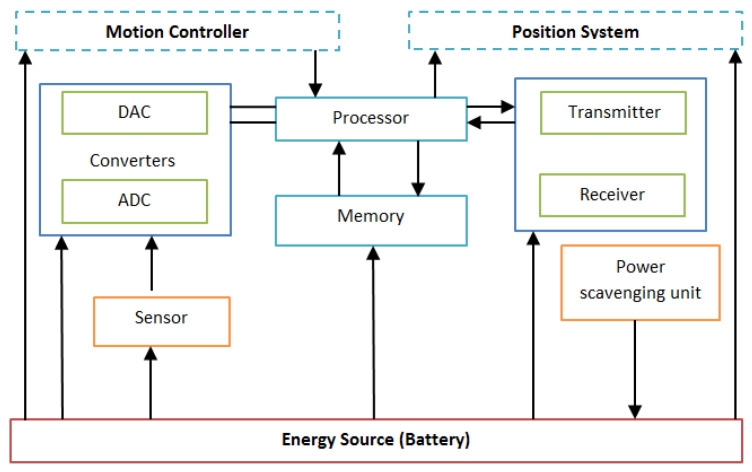
Generic sensor node structure.

**Figure 6 sensors-22-08279-f006:**
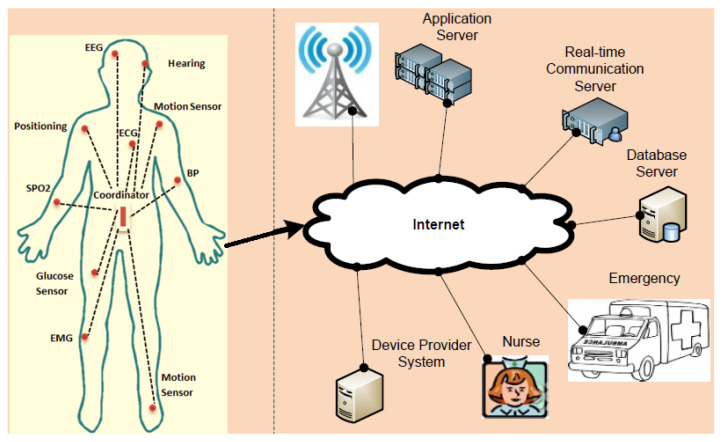
WBASN’s global connectivity.

**Figure 7 sensors-22-08279-f007:**
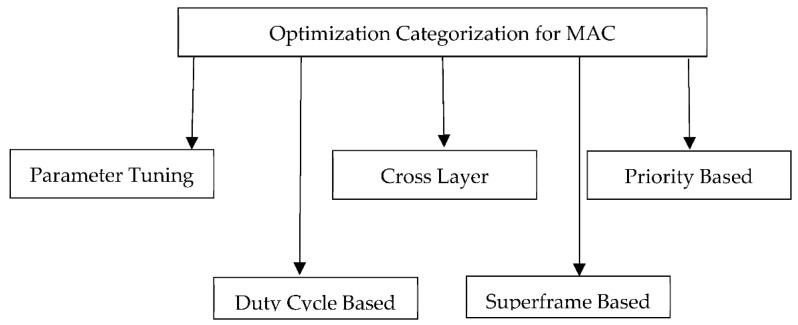
Optimisation approach categorisation for IEEE 802.15.4 and IEEE 802.15.6.

**Figure 8 sensors-22-08279-f008:**
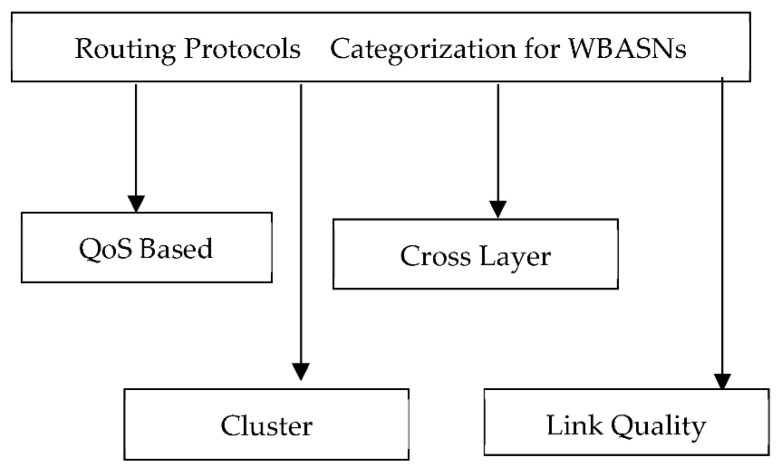
Categorisation of WBASN routing protocols.

**Table 1 sensors-22-08279-t001:** Requirements of sensors for patient monitoring systems [1].

Sensor Nodes	Data Generation Interval	Required Data Rate (Kbps)	Delay Requirement
ECG	4 ms	34	<125 ms
EMG	6 ms	19.6	<125 ms
EEG	4 ms	19.6	<125 ms
SpO2 (Pulse Oximeter)	10 ms	13.2	<250 ms
BP	10 ms	13.2	<250 ms
Respiration	40 ms	3.2	<250 ms
Skin temperature	60 s	2.27	<250 ms
Glucose sensor	250 s	0.528	<250 ms

**Table 2 sensors-22-08279-t002:** Characteristics and requirement analysis of WBASNs [55,56].

Parameters	Requirements
Lifetime	Long for wearable sensors and ultra-long for implanted sensors
Covered Area	Inside and around the body
Data Rate	Application dependent
Setup Time	Fast
Security	Simple and light mechanisms required
Customisation	Configurable sensor nodes
Fault Management	Detection mechanisms for the case of the node failure
Quality of Service	Application dependent
Power and Energy	Efficient energy and power mechanisms
Medium Access Control	Controllable, scalable and reliable
Frequency Bands	Medical bands and compatible with human tissues

**Table 3 sensors-22-08279-t003:** Chronic monitoring diseases with usable sensor types.

Diseases	Physiological Parameters	Biomedical Sensor Type
Cancer	Body fat sensor, weight loss indication sensor	Implantable/Wearable
Hypertension	BP	Implantable/Wearable
Heart Disease	ECG, BP, heart rate	Implantable/Wearable
Asthma	Respiration and oxygen saturation	Implantable/Wearable
Diabetes	Visual impairment	Wearable
Rheumatoid Arthritis	Joint stiffness	Wearable
Renal Failure	Urine output	Implantable
Vascular Diseases	blood pressure and peripheral perfusion	Implantable/Wearable
Infectious Diseases	Temperature	Wearable
Stroke	Activity recognition, impaired speech, memory etc.	Implantable/Wearable

**Table 4 sensors-22-08279-t004:** Frequency bands of IEEE 802.15.6.

Human-Body Communication	
Frequency	Bandwidth
16 MHz	4 MHz
27 MHz	4 MHz
Narrowband Communication	
Frequency	Bandwidth
402–405 MHz	300 KHz
420–450 MHz	300 KHz
863–870 MHz	400 KHz
902–928 MHz	500 KHz
956–956 MHz	400 KHz
2360–2400 MHz	1 MHz
2400–2438.5 MHz	1 MHz
UWB Communication	
13.2–4.7 GHz	499 MHz
6.2–10.3 GHz	z

**Table 5 sensors-22-08279-t005:** Characteristics and comparison of available technologies [67].

Technology	Data Rate	Frequency	Modulation	Channels	Topology	Range	Setup Time	Current Values	Market Adaptability for WBASNs
Bluetooth Classic	1–3 Mbps	2.4 GHz	GFSK	79	Scatternet	1–10 m	3 s	~45 mA	Low due to high power requirements
Bluetooth Low Energy	1 Mbps	2.4 GHz	GFSK	3	Piconet, Star	1–10 m	<100 s	~28 mA	Low due to power requirements and fewer channels
NB-IoT	234 Kbps	180 kHz	QPSK	13	Star	35 Km		120–300 mA	Low
LoRa (long range)	290 bps-50 Kbps	433 MHz, 868 MHz 915 MHz	SS chip	13 channels for 915 MHz	Star	10 Km		32 mA	Low as it is not open-source
IEEE 802.15.4(LRWPAN) /ZigBee	250 Kbps	2.4 GHz 868 MHz 915 MHz	O-QPSK	16	Star, Mesh	10–100 m	30 s	~16.5 mA	High for its suitability for wearable sensors in terms of QoS
IEEE 802.15.6	10 Kbps -10 Mbps	2.4 GHz, Narrowband HBC and UWB communication	D8PSK, DBPSK, DQPSK	Multiple channels according to frequency bands	Two hop Star, Mesh	1–5 m	<3 s	~1 mA	Still in the adoption stage as it also involves implanted sensors
ANT	1 Mbps	2.4 GHz	GFSK	125	Star, Mesh or tree	10–30 m		~22 mA	Low due to high power and limited QoS
Sensium	50 Kbps	868 MHz 915 MHz	BFSK	16	Star	1–5 m	<3 s	~3 mA	Low due to its low data rates
Zaralink ZL70101	50 Kbps	402–405 MHz 433–434 MHz	2FSK/4FSK	10	P2P	1–5 m	<3 s	~3 mA	Low due to its low data rates

**Table 6 sensors-22-08279-t006:** Comparison of WBASN standards with other wireless standards [1].

Standard	Provided Data Rate	Power Requirement	Battery Lifetime
WiFi	100 Mbps	100–1000 mW	Hours–days
Bluetooth	1–10 Mbps	4–100 mW	Days–weeks
Wibree	600 Kbps maximum	2–10 mW	Weeks–months
ZigBee	250 Kbps	3–10 mW	Weeks–months
802.15.4	250 Kbps maximum	3–10 mW	Weeks–months
802.15.6	1 Kbps–10 Mbps	0.1–2 mW	Months–years

**Table 7 sensors-22-08279-t007:** Comparative analysis of optimisation approaches.

MAC Optimisation Approaches	Advantages	Disadvantages
Parameter tuning	No explicit modification is required for IEEE 802.15.4 One-time parameters tuning is required Applicable to other standards, i.e., IEEE 802.15.6	Application-specific solutions Restricted to a theoretical range of parameters
Cross-layer	Optimal performance by using the information from other layers Adaptive to the situation	Overhead of the control messages High latency concerning medical applications
Duty-cycle-based	Adaptable with minimum modification to IEEE 802.15.4/ IEEE 802.15.6 Various opportunities to save power with the original standard	Add overhead to the coordinator for analysis and processing
Priority-based	Provide the required QoS to the transmitting nodes	Introduce operating and processing overheads
Superframe modification	Provide scalability, multiple topologies support, make IEEE 802.15.4/ IEEE 802.15.6 more adaptive	Major changes in operations of the standard Adaptability demands resource usage for sensor and coordinator nodes

**Table 8 sensors-22-08279-t008:** Comparison of WBASN MAC protocols.

Protocol	Year	Standard	Access scheme	Shortcomings	QoS
DQBAN [100]	2009	IEEE 802.15.4	Hybrid	Requires the management of different queues as well as fuzzy-logic system implementation in every sensor node	R, C
EELDC [101]	2009	IEEE 802.15.4	TDMA	Fixed scheduling is used for data transmission, which does not fulfil the application diversity in WBASNs	E, R
BDD [102]	2009	IEEE 802.15.4	TDMA	The performance is only validated for one biomedical sensor, i.e., ECG; hence, QoS performance in a scalable environment is a concern	E
U-MAC [103]	2010	IEEE 802.15.4	Slotted ALOHA	Complex and involve overheads in terms of data categorisation and identification of retransmission packets	D
HUA-MAC [104]	2010	IEEE 802.15.4	Slotted ALOHA	Shows QoS limitations in the scalable and diverse application scenarios	D, R
PNP-MAC [105]	2010	IEEE 802.15.4	Hybrid	The traffic loads of low-priority biomedical sensors are ignored, which may cause delay and consume more energy in the case of retransmission	D, E
CA-MAC [106]	2011	IEEE 802.15.4	Hybrid	Dynamic change in the frame structure, which is not easy to implement with the IEEE 802.15.4/IEEE802.15.6 standard	R
LDTA-MAC [58]	2011	IEEE 802.15.4	Hybrid	Successful execution of such protocol requires a good synchronisation mechanism between node and superframe; moreover, a clear priority assignment scheme is missing	D
MEB-MAC [107]	2012	IEEE 802.15.6	Hybrid	Scalability is a concern as the insertion of many new slots will create QoS degradation for the other nodes of the network	D
D2MAC [108]	2013	IEEE 802.15.4	Slotted CSMA/CA	Consideration of single QoS parameters from the application, i.e., data rates to make the protocol adaptive	D
EMAC [109]	2013	IEEE 802.15.4	Hybrid	The channel characterisation and integration issues of these relay nodes are not discussed, which is an important aspect in validating performance	E
C-MAC [110]	2013	IEEE 802.15.6	TDMA-FDMA	The solution is complex due to the usage of multiple access mechanisms simultaneously, i.e., TDMA and FDMA; strong synchronisation is needed	C, M
ATLAS [99]	2013	IEEE 802.15.4	Hybrid	A detailed discussion about the backoff procedure for the waiting nodes in this modified scheme is missing; moreover, adding an additional mechanism on IEEE 802.15.4 may cause more energy consumption for sensor nodes	P
PLA-MAC [111]	2013	IEEE 802.15.4	Hybrid	To adopt this mechanism, more energy sources are required, whereas energy efficiency computation is not discussed in the simulations	P, R
Single-radio multi-channel TDMA MAC protocol [112]	2014	IEEE 802.15.4	TDMA	The management of multi-channels is still challenging due to co-channel interference and restricted band allocation	D
MFS-MAC [49]	2014	IEEE 802.15.6	Hybrid	There is a need to define the authorities of the master node; moreover, this solution is not scalable	E
PMAC [113]	2014	IEEE 802.15.4	Hybrid	The applied security mechanism requires more time for sharing key and decryption, which can hinder the effectiveness of this protocol in terms of stringent QoS for WBASNs	P, S
HEH-BMAC [114]	2015	IEEE 802.15.4	Hybrid	Its suitability for critical medical applications is not discussed, whereas such applications require limited latency and high reliability	P, E
RC-MAC [115]	2015	IEEE 802.15.4	Hybrid	Receiver centric access mechanism demands resources in terms of power; moreover, the synchronisation among receiving nodes to avoid collision exploits the duty cycle mechanism	T
PA-MAC [116]	2016	IEEE 802.15.4 IEEE 802.15.6	Hybrid	It requires hardware modification, which is a difficult task for existing standards	P, E, C
AT-MAC [117]	2016	IEEE 802.15.4	Hybrid	The proposed mechanism focuses on reliability for WBASN medical applications, whereas a trade-off discussion between reliability, delay and energy usage is missing	R
CoR-MAC [118]	2016	IEEE 802.15.4, IEEE 802.15.6	Hybrid	For the implementation of such a mechanism, strong synchronisation is required between reservation mechanisms, which require more processing power and memory	D
C-MAC+ [110]	2017	IEEE 802.15.6	Hybrid	A strong a-synchronisation mechanism is required to avoid collision by incorporating a duty cycle mechanism. An extensive modification is required to implement C-MAC in existing standards	D, E
Interference mitigation model [119]	2018	IEEE 802.15.6	CSMA/CA	Required more resources in terms of energy and memory due to queue management	M, T
TCP-CSMA/CA [120]	2019	IEEE 802.15.4	Slotted CSMA/CA	Implementation requires more energy consumption and could add more delays for not-prioritised traffic	P, D
TA-MAC [121]	2019	IEEE 802.15.4	Hybrid	The proposed traffic-based priority mechanism works well; however, inclined average delay values for the other traffic types are noticed	P
DCSS [122]	2019	IEEE 802.15.6	Hybrid	The proposed dynamic channel selection mechanism selects a good channel to avoid interference; however, for that, it needs information from the physical layer, which will require more time and resources	I, T
PBDT [123]	2019	IEEE 802.15.6	Hybrid	Posture-based data transmission helps to identify the posture based on RSSI values; however, the proposed mechanism is complex and maybe not be suitable for sensors with delay-sensitive data	I, M

**Table 9 sensors-22-08279-t009:** Comparative analysis of QoS-based protocols.

Protocols	Comparison Parameters	
	QoS Focus	Methodology
QPRR [56]	Reliability	Link reliability using EWMA Use of RSSI for localisation Numerical modelling
QPRD [132]	Delay	Queuing and channel delay using EWMA Use of RSSI for localisation
DMQoS [133]	Delay, reliability, priority traffic	Lexicographic optimisation for energy-aware forwarding, Greedy approach for reliability Queuing delay using EWMA, transmission delay using weighted average transmission delay (WATD) Link reliability using windowed mean EWMA (WMEWMA)
LOCALMOR [134]	Latency, energy reliability, priority traffic, residual	Power efficiency module uses a min–max approach Little’s formula for queuing delay
RL-QRP [135]	Packet delivery, delay, congestion	Q-value implementations
EN-NEAT [136]	Energy, packet delivery	EN-NEAT utilises multi-hop communication to reduce energy depletion and maximises network longevity. Firstly, avoid the transmission of normal data. Secondly, compare the sensed information, and if there is a variation, a transmission occurs; otherwise, no transmission occurs. Lastly, a minimum cost function was proposed to carefully choose the parent or forwarder node that has the highest residual energy and the shortest distance to sink.
Temperature-aware routing [137]	Energy, packet delivery, Delay	In the proposed work, a secondary base station is selected; this helps to reduce the temperature of the neighbour nodes, as these neighbour nodes will not be a part of the new data routes. At this time, the sensor node will transmit only the priority packets to the secondary base stations.
TARA [138]	Energy, priority and throughput	A thermal-aware routing algorithm is proposed to reduce the number of transmissions from hot-spot nodes or the nodes bearing more traffic by assigning the priorities

**Table 10 sensors-22-08279-t010:** Comparative analysis of cross-layer protocols.

Protocols	Comparison Parameters
	Methodology
WASP [139]	The distributed tree is used for channel access and multi-hop routing Parent and child nodes share information
CICADA [140] CICADA-S	A distributed approach using a tree-based algorithm with a TDMA-based mechanism
TICOSS [141]	Multi-hop communication by dividing the network into time zones in IEEE 802.15.4 V-scheduling is used for collision avoidance
BIOCOMM [142] BIOCOMM-D	MAC layer and routing layer coordination of neighbour tables; the MAC layer keeps updating the routing layer for neighbour status
Tree-based energy-efficient routing [143]	Tree-based approach Provides better energy consumption in two ways: (a) establishing an energy-efficient end-to-end path, (b) adaptive transmission power mechanism for the nodes according to distance
Optimising transmission reliability, energy efficiency, and lifetime in body sensor networks [144]	Proposed a cross-layer energy-efficient algorithm that utilises different characteristics of different layers, including the physical layer, media access control (MAC) and the network layer The proposed structure also uses optimal power control on a single link to reduce power consumption, which, in turn, prolongs the overall network lifetime
Thermal-aware routing protocol [145]	To control the temperature of the sensor nodes, two thresholds are defined for the avoidance and recovery of heat-up devices. Once these thresholds are reached, the node is declared a hot-spot node and its usage is temporarily blocked. After the cooling procedure, the node will once again participate in the data routing procedure.

**Table 11 sensors-22-08279-t011:** Comparative analysis of cluster protocols.

Protocols	Comparison Parameters
	Methods
AnyBody [146]	Overall, the protocol works in five steps, i.e., density computation, cluster-head construction, backbone network setup, routing path setup and neighbour discovery
LEACH [147]	The cluster heads aggregate and compress the data and forward it to the base station Each node uses a stochastic algorithm at each round to determine whether it will become a cluster head in this round
HIT [148]	To minimise energy consumption, parallel processing is used for intra-cluster and inter-cluster communication modes. At the start, one node is selected as the cluster head, and later, further cluster heads are selected. Time division multiple access (TDMA) scheduling is used in HIT to send data to upstream and downstream nodes
LEACH-M [105]	LEACH-M supports sensor nodes’ mobility in WSNs by adding membership declarations to the LEACH protocol LEACH-Mobile outperforms LEACH in terms of packet loss in a mobility-centric environment
LEACH-EE [109]	It prolongs the network lifetime and reduces energy consumption by first gathering data by cluster head; then, an optimal multi-hop path that leads to the base station is formed among the cluster heads. In this way, the problem of cluster heads consuming more energy is solved.
AZM-LEACH [110]	Improved version of BIOCOMM providing better delay performance in the multi-hop scenario.
LEACH-GA [107]	The ant colony approach is used. The core idea is to evaluate cluster heads’ current residual energy and location information from the perspective of the overall network in real-time; a single ant traverses all nodes at once, forming a dendritic multi-hop path. While the path is rebuilt, low-energy nodes select an energy-saving path, and high-energy nodes increase energy consumption as a consideration to prolong the network lifetime.
LEACH-IACA [149]	A multi-hop routing algorithm LEACH-GA (LEACH-Genetic Algorithm) improves the cluster heads’ single-hop system in the LEACH routing protocol of heterogeneous wireless sensor networks. In this protocol, the cluster heads provide the shortest path link with SINK. The cluster heads that are far away from SINK communicate with SINK through the transit cluster heads. In fact, those who are near SINK can communicate with it directly. From this point, the LEACH algorithm can be improved to be an algorithm with a SINK-centred multi-hop tree cluster link.
EB-MADM [150]	The energy budget based multiple attribute decision making (EB-MADM) algorithm for cooperative clustering is used for dynamic cluster selection. Provides energy efficiency.
BAN-Trust [151]	An attack-resilient malicious node detection scheme (BAN-Trust) is brought into the current system; it can identify malignant attacks on BANs. In this BAN-Trust scheme, malignant nodes are identified according to the nature acquired through the nodes on their own and the approvals shared by various nodes.

## Data Availability

Not applicable.

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
