# Peer review of "Wireless Body Area Sensor Networks: Survey of MAC and Routing Protocols for Patient Monitoring under IEEE 802.15.4 and IEEE 802.15.6"

_sensors, 2022, doi:10.3390/s22218279_

Round 1

Reviewer 1 Report

It can be clearly seen that in this review, techniques for media access control (MAC) and routing protocols are investigated with regard to the application needs of patient monitoring systems. In addition, the authors discuss the open issues of the communication protocol layer, trying to provide guidelines for the design and development of the WBASN field.

In general, the structure of the paper is reasonable and the content is relatively rich. But there is an important problem, the work summarized in the article is up to 2019, and it seems a little out of date for a review article trying to publish in 2021. It is hoped that the authors will add some recent developments to enhance the value of the paper.

Author Response

We thank the reviewer for the encouraging comments.

We have now included the following recent articles in the revised versions.

  1. Hajar, Muhammad Shadi, M. Omar Al-Kadri, and Harsha Kumara Kalutarage. "A survey on wireless body area networks: Architecture, security challenges and research opportunities." Computers & Security 104 (2021): 102211.
  2. Liu, Qingling, Kefa G. Mkongwa, and Chaozhu Zhang. "Performance issues in wireless body area networks for the healthcare application: a survey and future prospects." SN Applied Sciences 3, no. 2 (2021): 1-19.
  3. Zhang, Kai, Ping Jack Soh, and Sen Yan. "Meta-wearable antennas—a review of metamaterial based antennas in wireless body area networks." Materials 14, no. 1 (2020): 149.
  4. Fotouhi, Mahdi, Majid Bayat, Ashok Kumar Das, Hossein Abdi Nasib Far, S. Morteza Pournaghi, and Mohammad-Ali Doostari. "A lightweight and secure two-factor authentication scheme for wireless body area networks in health-care IoT." Computer Networks 177 (2020): 107333.
  5. Asam, Muhammad, Tauseef Jamal, Muhammad Adeel, Areeb Hassan, Shariq Aziz Butt, Aleena Ajaz, and Maryam Gulzar. "Challenges in wireless body area network." International Journal of Advanced Computer Science and Applications 10, no. 11 (2019).
  6. Anguraj, Dinesh Kumar, and S. Smys. "Trust-based intrusion detection and clustering approach for wireless body area networks." Wireless Personal Communications 104, no. 1 (2019): 1-20.

Reviewer 2 Report

In the proposed manuscript, the authors are presenting the survey of wireless body area sensor networks under communications standards like IEEE 802.15.4 and IEEE 802.15.6. The primary use of such wireless body area sensor networks is various patient monitoring scenarios. The focus is on reviewing MAC (medium access control) and routing protocols in sense of application requirements of patient monitoring systems.

The manuscript is, in general, well-organized and well-written. The language is good, and the text reads fluently. Although the manuscript is a review-type of paper, the number of references is on the upper limit.

Comments:

References: reference no. 3 is not referred to in the text. Please make sure that all the references are properly mentioned in the manuscript. The references are also not in the same sequence as the referencing order in the manuscript.

Figure 4.: Figure 4 shows a WBASN sensor node structure. I would suggest that the caption would describe it as a generic node structure because different nodes can have different structures, but some of the structures can be common, therefore, generic.

Line no. 203-204: Various companies make processors for WBASNs,… I doubt that any “processor” (or, better said, microcontroller) is made specifically for WBASN. It would be better that microprocessors and microcontrollers with specific properties that fit the WBASN use case are presented (or the needed properties).

Table 5: It is important that for the individual technology, the typical power supply voltage is also stated beside the current consumption so a reader can calculate the power consumption, or better, the table presents the consumption in mW (milliwatts). There is also an error in reference no. [58] says that the technology is Sensium, not Senium as it is in Table 5 (8th row).

Subchapter 2.7. Power consumption: It should be explained that data rate and communication protocol carrier frequency have a huge impact on power requirements/consumption. Higher frequency + higher data rate means higher power consumption.

Although there is some conclusion at the end but regards the extensive number of comparisons in the manuscript and references, some deepened discussion at the end could be provided.

Author Response

Comment 1: Reference no. 3 is not referred to in the text. Please make sure that all the references are properly mentioned in the manuscript. The references are also not in the same sequence as the referencing order in the manuscript.

Response: We thank the reviewer for pointing this out. We have now referred to it on page 4 line 41 in the revised version. We have also made sure that the order of the reference is correct in the revised version.

Comment 2: Figure 4 shows a WBASN sensor node structure. I would suggest that the caption would describe it as a generic node structure because different nodes can have different structures, but some of the structures can be common, therefore, generic.

Response: We thank the reviewer for this valuable suggestion. We have now changed the caption for figure 4 in the revised version.

Comment 3: Line no. 203-204: Various companies make processors for WBASNs,… I doubt that any “processor” (or, better said, microcontroller) is made specifically for WBASN. It would be better that microprocessors and microcontrollers with specific properties that fit the WBASN use case are presented (or the needed properties).

Response: We thank the reviewer for this insightful comment. We have now replaced the word “processor” with “microcontroller” as suggested by the respected reviewer.

Comment 4: Table 5: It is important that for the individual technology, the typical power supply voltage is also stated beside the current consumption so a reader can calculate the power consumption, or better, the table presents the consumption in mW (milliwatts). There is also an error in reference no. [58] says that the technology is Sensium, not Senium as it is in Table 5 (8th row).

Response: We thank the reviewer for this comment. We agree with the reviewer that adding such information would be beneficial for the users, however, these details were missing from the reviewed articles and hence are not included here. We have now made the correction in table 5 row 8.

Comment 5: Subchapter 2.7. Power consumption: It should be explained that data rate and communication protocol carrier frequency have a huge impact on power requirements/consumption. Higher frequency + higher data rate means higher power consumption.

Response: We thank the reviewer for the suggestion. We have now added this explanation on page 5 lines 371-373.

Comment 6: Although there is some conclusion at the end but regards the extensive number of comparisons in the manuscript and references, some deepened discussion at the end could be provided.

Response: We thank the reviewer for this constructive comment. We have updated our conclusion in the revised version and have added more details about the reviewed work.

Reviewer 3 Report

References are TOO Much

References are outdated (no References in 2020,2021, and 2022)

A flow chart or Added taxonomy will be essential to review the whole research efforts and possible modeling techniques in previous works.

The paper objectives must be correlated to 6G and its main features especially the number of sensors and required delay constraints.

Author Response

Comment 1: References are too much.

Response: We thank the reviewer for bringing up this important point. We understand that we have included a lot of references, however, all of these references are related to our work and add value to the article. We believe that keeping these references will help the reader in getting a comprehensive overview of this area.  

Comment 2: References are outdated (no References in 2020,2021, and 2022).

Response: We thank the reviewer for this constructive comment. We have now included the following recent articles in the revised version.

  1. Hajar, Muhammad Shadi, M. Omar Al-Kadri, and Harsha Kumara Kalutarage. "A survey on wireless body area networks: Architecture, security challenges and research opportunities." Computers & Security 104 (2021): 102211.
  2. Liu, Qingling, Kefa G. Mkongwa, and Chaozhu Zhang. "Performance issues in wireless body area networks for the healthcare application: a survey and future prospects." SN Applied Sciences 3, no. 2 (2021): 1-19.
  3. Zhang, Kai, Ping Jack Soh, and Sen Yan. "Meta-wearable antennas—a review of metamaterial based antennas in wireless body area networks." Materials 14, no. 1 (2020): 149.
  4. Fotouhi, Mahdi, Majid Bayat, Ashok Kumar Das, Hossein Abdi Nasib Far, S. Morteza Pournaghi, and Mohammad-Ali Doostari. "A lightweight and secure two-factor authentication scheme for wireless body area networks in health-care IoT." Computer Networks 177 (2020): 107333.
  5. Asam, Muhammad, Tauseef Jamal, Muhammad Adeel, Areeb Hassan, Shariq Aziz Butt, Aleena Ajaz, and Maryam Gulzar. "Challenges in wireless body area network." International Journal of Advanced Computer Science and Applications 10, no. 11 (2019).
  6. Anguraj, Dinesh Kumar, and S. Smys. "Trust-based intrusion detection and clustering approach for wireless body area networks." Wireless Personal Communications 104, no. 1 (2019): 1-20.

Comment 3: A flow chart or Added taxonomy will be essential to review the whole research efforts and possible modelling techniques in previous works.

Response: We thank the reviewer for this valuable comment. We have added a taxonomy in Figure 2 on page 5 which provides an overview of the research efforts presented.

Comment 4: The paper objectives must be correlated to 6G and its main features especially the number of sensors and required delay constraints.

Response: We thank the reviewer for this comment. We have revised the paper objectives and have included the suggested details in the revised version.

Round 2

Reviewer 1 Report

The author has revised my concerns.